# Estimating Bus Mass Using a Hybrid Approach: Integrating Forgetting Factor Recursive Least Squares with the Extended Kalman Filter

**DOI:** 10.3390/s25061741

**Published:** 2025-03-11

**Authors:** Jingyang Du, Qian Wang, Xiaolei Yuan

**Affiliations:** School of Automobile, Chang’an University, Xi’an 710064, China; 2022900653@chd.edu.cn (J.D.); 2023900815@chd.edu.cn (Q.W.)

**Keywords:** mass estimation of bus, Robust FFRLS, EKF, low-cost solution

## Abstract

The vehicle mass is a crucial state variable for achieving safe and energy-efficient driving, as it directly impacts the vehicle’s power performance, braking efficiency, and handling stability. However, current methods frequently rely on particular operating conditions or supplementary sensors, which limits their ability to provide accurate, stable, and convenient vehicle mass estimation. Moreover, as a form of public transportation, buses are subject to stringent safety standards. The frequent variations in passenger numbers result in substantial fluctuations in vehicle mass, thereby complicating the accuracy of mass estimation. To address these challenges, this paper proposes a hybrid vehicle mass estimation algorithm that integrates Robust Forgetting Factor Recursive Least Squares (Robust FFRLS) and Extended Kalman Filter (EKF). By sequentially employing these two methods, the algorithm conducts dual-stage mass estimation and incorporates a proportional coordination factor to balance the outputs from FFRLS and EKF, thereby improving the accuracy of the estimated mass. Importantly, the proposed method does not necessitate the installation of new sensors, relying instead on data from existing CAN-bus and IMU sensors, thus addressing cost control concerns for mass-produced vehicles. The algorithm was validated through MATLAB(2022b)-TruckSim(2019.0) simulations under three loading conditions: empty, half-load, and full-load. The results demonstrate that the proposed algorithm maintains an error rate below 10% across all conditions, outperforming single-method approaches and meeting the stringent requirements for vehicle mass estimation in safety and stability functions. Future work will focus on conducting real-world tests under various driving conditions to further validate the robustness and applicability of the proposed method.

## 1. Introduction

With the rapid development of electric and intelligent vehicles, the safety and energy efficiency of cars have been greatly improved, effectively reducing the incidence of traffic accidents and lowering fuel consumption and pollution emissions [1]. To achieve higher levels of automotive active safety and intelligent control, precise estimation of the vehicle’s own state and motion state is required [2]. Among the factors, the total vehicle mass is a critical one. For large buses, these vehicles exhibit characteristics such as a high center of gravity, heavy payload, complex operating conditions, and significant fluctuations in occupancy during operation. The total vehicle mass varies considerably between empty and full loaded states, which must be carefully considered in the design of the control system [3]. Therefore, achieving precise, stable, and robust estimation of the total vehicle mass is of great significance for the safety and intelligent control of vehicles.

For the estimation of the total vehicle mass, a straightforward approach is to measure it by introducing new sensors. The total vehicle mass can be indirectly acquired by measuring the shape variables of the load-bearing parts of the vehicle under different load conditions by sensors [4]. For instance, variations in tire pressure can be measured using tire pressure sensors [5], and the deformation extent of the suspension can be measured with strain gauge sensors [4], thereby obtaining the total vehicle mass. However, the method of measurement through sensor installation has significant limitations in equipment installation, measurement environment, and data processing [6]. Furthermore, for original equipment manufacturers (OEMs), the addition of new sensors will lead to an increase in costs, which, from an economic cost perspective, also restricts the large-scale application of this method. Currently, it is only supported for limited use in specific scenarios.

The vehicle mass, as a critical parameter in automotive dynamics, has received increasing attention in recent years owing to advancements in methodologies and technologies. Current approaches for estimating vehicle mass can be broadly classified into two categories: based on vehicle dynamics models and deep learning techniques [7].

Methods based on vehicle dynamics models are widely employed to study the longitudinal dynamics of vehicles. In recent years, with the advancement of intelligent technologies in commercial vehicles, IMU sensors have been increasingly utilized in both passenger and commercial vehicles, significantly enhancing the accuracy of overall vehicle mass estimation. From a methodological perspective, the primary approaches for estimating vehicle mass using dynamics models include recursive least squares (RLS) [8,9] and Kalman filtering (KF) algorithms [10,11]. Vahidi et al. [12] introduced an RLS method with multiple forgetting factors, accounting for varying rates of parameter changes, to achieve segment-wise stable estimates of vehicle mass. To address the significant impact of pulse interference (such as braking and gear shifting) on the accuracy of RLS-based methods, Chor et al. [13] proposed a robust multi-forgetting factor recursive least M-squared algorithm, which ensures reliable mass estimation even under pulse interference conditions and remains accurate at low sampling frequencies. Additionally, to mitigate sensor noise and leverage data from multiple integrated sensors, Kalman filtering algorithms are extensively applied for vehicle mass estimation. SUN et al. [14] proposed a dual-layer estimation framework for electric buses based on their operational characteristics. This framework first estimates road slope using the Extended Kalman Filter (EKF) algorithm, and subsequently employs RLS and EKF algorithms to simultaneously estimate vehicle mass. The estimated results are then combined with weighting. Yang et al. [15], leveraging the explicit correlation between road roughness and the suspension system, introduced an adaptive Extended Kalman Filter–unified inverse (AEKF-UI) algorithm that can concurrently estimate vehicle mass changes and unknown road roughness inputs. Zhang et al. [16] developed a dual robust embedded cubic Kalman filter (RECKF) algorithm, which accounts for unknown measurement noise, for joint estimation of vehicle mass and center of gravity, thereby significantly improving estimation accuracy.

In recent years, with the rapid advancement of artificial intelligence technology and its widespread application across various industries, deep learning has emerged as a powerful tool for state estimation due to its ability to model complex nonlinear systems. Vehicle dynamics, being inherently nonlinear, presents significant challenges for accurate state estimation using traditional dynamic methods. Korayem et al. [17] addressed this challenge by analyzing the system and selecting appropriate feature parameters. They employed a deep neural network (DNN) approach to estimate the total vehicle mass, achieving an estimation error of less than 10%. The results were compared with those obtained from dynamic methods, demonstrating the superiority of the deep learning-based approach. Zhang et al. [18] proposed a dynamic total vehicle mass estimation framework based on bidirectional gated recurrent units (BiGRUs), which significantly improved the accuracy of deep neural network estimations. However, deep learning methods require substantial amounts of data for training, which can be prohibitively expensive for applications involving buses. Consequently, there has been a growing trend toward integrating dynamic model-based methods with deep learning techniques for state estimation. Jin et al. [8] combined the square-root cubature Kalman filter (SCKF) method based on longitudinal dynamics with the BiLSTM method from deep learning, thereby enhancing both the accuracy and stability of total vehicle mass estimation. Zhang et al. [19] initially used a feedforward neural network (FFNN) to learn the correlations between total vehicle mass and other state parameters such as longitudinal speed and acceleration. Subsequently, they applied the recursive least squares (RLS) algorithm for mass estimation and designed a fuzzy logic system to integrate the two methods. This hybrid approach exhibited superior robustness and stability.

Despite the significant achievements of both dynamic-based and deep-learning-based methods in vehicle mass estimation, each approach has its limitations and can exhibit high error rates under certain conditions. For example, during normal driving scenarios such as braking and gear shifting, the dynamic model may become misaligned, leading to substantial inaccuracies in mass estimation [20]. To address this issue, it is common to impose strict initialization conditions on the algorithms; however, adding too many constraints can slow down convergence and hinder real-time estimation capabilities [21]. Furthermore, non-Gaussian noise can severely disrupt dynamic-based methods, resulting in larger estimation errors [22]. While deep-learning-based methods are powerful, they require extensive datasets for training. Collecting such data must consider various factors, including different load distributions, driving behaviors, and environmental conditions, which poses significant challenges [23]. Additionally, when encountering unknown or unseen data, deep-learning models may experience rapid degradation in estimation accuracy, leading to instability in the results, which can critically impact driving safety.

To address the aforementioned challenges, this paper proposes a hybrid estimation method that integrates the EKF with Robust Forgetting Factor Recursive Least Squares (Robust FFRLS), focusing on buses as the research object. Initially, the total vehicle mass is estimated independently using EKF and Robust FFRLS. Subsequently, a weighted hybrid approach is employed to combine the outputs from both methods, leveraging their respective advantages under different operating conditions to achieve faster convergence and lower error rates. Finally, the proposed algorithm is validated in a simulation environment. Importantly, all sensor data required for this method can be obtained from existing vehicle sensors, eliminating the need for additional hardware and providing a cost-effective solution for whole-vehicle mass estimation.

The structure of this paper is organized as follows. Section 2 describes the modeling process of the bus dynamic model. Section 3 outlines the hybrid estimation framework, detailing the derivation of the total vehicle mass estimation using EKF and Robust FFRLS, followed by an explanation of the weighted hybrid method. Section 4 presents the validation of the hybrid algorithm under three typical loading conditions: empty, half-loaded, and fully loaded.

## 2. Bus Model Building

### 2.1. Bus Longitudinal Dynamics Model

During operation, buses primarily operate on structured roads such as urban streets and highways. The longitudinal dynamic model of a large bus, constructed based on Newton’s second law, is illustrated in Figure 1. In this model, the driving force propels the vehicle forward by overcoming four main resistances: air resistance, rolling resistance, grade resistance, and inertial resistance due to acceleration.

Aerodynamic resistance arises from the pressure and frictional forces exerted on the vehicle by the air during operation. Rolling resistance originates from the deformation and friction between the tires and the road surface when they are in contact. Grade resistance is caused by the component of gravitational force acting on the vehicle when it travels uphill or downhill. The longitudinal dynamics equation of the vehicle can be expressed as follows:(1)$$M{\dot{v}}_{x}=F_{t}-(F_{w}+F_{i}+F_{f})$$
where *M* represents the vehicle mass, ${\dot{v}}_{x}$ represents the longitudinal acceleration, $F_{t}$ represents the driving force, $F_{w}$ represents the air resistance, $F_{i}$ represents the grade resistance, and $F_{f}$ represents the rolling resistance.

The driving force is obtained by calculating the engine torque $T_{tq}$ transmitted through the power transmission system to the driven wheels. The formula is:(2)$$F_{t}=\frac{T_{tq}i_{g}i_{0}\eta_{T}}{r}$$
where $T_{tq}$ represents the engine torque transmitted to the wheels, $i_{g}$ is the gearbox transmission ratio, $i_{0}$ is the transmission ratio of the main reduction gear, $\eta_{T}$ is the mechanical efficiency of the transmission system, and *r* is the rolling radius of the tires. $T_{tq}$ and $i_{g}$ can be obtained through the CAN bus signal.

If the vehicle is traveling under no wind conditions, then the air resistance $F_{w}$ is:(3)$$F_{w}=\frac{1}{2}C_{D}\rho Av_{x}^{2}$$
where $C_{D}$ is the coefficient of air resistance, ρ is the density of air, $v_{x}$ is the traveling speed, and A is the frontal area, which is the projected area of the vehicle in the direction of travel. $v_{x}$ can be obtained through the CAN bus signal.

When driving uphill, the grade resistance force $F_{i}$ that the vehicle experiences is the component of its weight force along the slope direction:(4)$$F_{i}=Mgsin\alpha$$

The rolling resistance coefficient f of a vehicle’s tire is approximately linearly related to the vehicle speed $v_{x}$, and f can be expressed as $f=f_{0}+f_{1}v_{x}$. Therefore, the rolling resistance $F_{f}$ is:(5)$$F_{f}=Mg\left(f_{0}+f_{1}v_{x}\right)\cos\alpha$$

In this equation, g is the gravitational acceleration, α is the slope angle, $f_{0}$ and $f_{1}$ are the speed-dependent coefficients of rolling resistance, $f_{0}$ is the constant term and $f_{1}$ is the coefficient of the first-order term, and M is the mass, including the vehicle’s loaded mass $M_{Z}$ and the vehicle’s curb mass $M_{C}$.

In fact, according to the road grade index specified in the reference standard for highway route design, the maximum grade of a road is 8%. Since the road grade for bus operation is generally smaller, it can be assumed that:(6)$$\left\{\begin{array}{c} \sin\alpha \approx \tan\alpha =\alpha \\ \cos\alpha \approx 1 \end{array}\right.$$

Then the grade resistance $F_{i}$ and rolling resistance $F_{f}$ are:(7)$$\left\{\begin{array}{c} F_{i}=Mg\alpha \\ F_{f}=Mg\left(f_{0}+f_{1}v_{x}\right) \end{array}\right.$$

By decomposing the various forces, Equation (1) can be further expressed as:(8)$$M{\dot{v}}_{x}=\frac{T_{tq}i_{g}i_{0}\eta_{T}}{r}-\left[\frac{1}{2}C_{D}\rho Av_{x}^{2}+Mg\alpha +Mg\left(f_{0}+f_{1}v_{x}\right)\right]$$

### 2.2. Kinematic Model Based on IMU Sensor

With the advancement of automotive intelligence, IMU and other sensors have been extensively deployed to accurately measure the vehicle’s acceleration along three axes and angular velocity about these axes [24]. When the vehicle is driving uphill, the acceleration sensor measures values of $a_{senx}$ on the X axis and $a_{senz}$ on the Z axis. The measured value of $a_{senx}$ by the acceleration sensor is not only related to the instantaneous driving acceleration ${\dot{v}}_{x}$, but also affected by the current slope angle α. $a_{senx}$ is the sum of the vehicle’s driving acceleration ${\dot{v}}_{x}$, and the component of gravity acceleration g along the slope gsinα. If we consider cornering conditions, the following equation applies:(9)$$a_{senx}=g\sin\alpha +{\dot{v}}_{x}-\dot{\varphi }v_{y}$$

In this equation, $a_{senx}$ is the longitudinal acceleration measured by the IMU sensor, $v_{y}$ is the lateral velocity, and $\dot{\varphi }$ is the yaw rate.

$a_{senz}$ is the component of gravity acceleration g that is perpendicular to the slope:(10)$$a_{senz}=g\cos\alpha$$

When the vehicle is traveling in a straight line, Equation (9) can be further simplified as:(11)$$a_{senx}=g\sin\alpha +{\dot{v}}_{x}$$

Further, by combining the longitudinal kinematic model based on IMU sensors with the longitudinal dynamic model, i.e., by combining Equation (8) with Equation (11), and assuming a small slope assumption, the longitudinal dynamic expression based on velocity sensor can be obtained as follows:(12)$$Ma_{senx}=\frac{T_{tq}i_{g}i_{0}\eta_{T}}{r}-\left[\frac{1}{2}C_{D}\rho Av^{2}+Mg\alpha +Mg\left(f_{0}+f_{1}v\right)\right]$$

## 3. Vehicle Mass Estimation Approach Using a Hybrid Algorithm Combining Variable-Structure EKF and Robust FFRLS

### 3.1. Mass Estimation Based on Robust FFRLS

The RLS algorithm, based on the principle of least squares, recursively updates parameters to adapt to changes in input signals in real time. Due to its fast convergence rate and superior performance, it is widely employed for identifying vehicle state parameters. The FFRLS algorithm enhances the traditional RLS by incorporating a forgetting factor, which assigns greater weight to new data compared to older data. This modification addresses the issue of excessive reliance on historical data in traditional RLS algorithms, thereby improving adaptability in non-stationary environments [25]. Given that buses operate in dynamic and non-stationary conditions, employing FFRLS for estimating the total vehicle mass can more rapidly accommodate new data and information, ensuring reliable performance. The standard form of the FFRLS formula can be expressed as:(13)$$y=\varphi^{T}\theta +e$$

Combining Equation (13) with Equation (12), the individual parameters can be expressed specifically as follows:(14)$$\left\{\begin{array}{c} y=T_{tq}i_{g}i_{0}\eta_{T}/r-0.5C_{D}\rho A{\nu_{x}}^{2}\\ \varphi =a_{senx}+(f_{0}+f_{1}v_{x}+\alpha )g\\ \theta =M \end{array}\right.$$

In the above equation: y is the output variable, φ is the measurable data, θ is the identification parameter, and e is Gaussian white noise.

Considering that RLS uses the minimization of residual squared error, it is sensitive to abnormal measurement data [26], abnormal measurement data input will have a significant impact on the accuracy of the overall vehicle mass estimation. In order to reduce the influence of sensor abnormal error on the accuracy of the overall vehicle mass estimation, the idea of the Huber function is introduced into FFRLS to enhance the robustness of the algorithm. The standard form of FFRLS after time discretization is as follows:(15)$$\left\{\begin{array}{c} {\hat{\theta }}_{k}={\hat{\theta }}_{k-1}+K_{k}(y_{k}-\varphi_{k}^{T}{\hat{\theta }}_{k-1})\\ K_{k}=P_{k-1}\varphi_{k}^{T}{\left(\lambda_{k}+\varphi_{k}^{T}P_{k-1}\varphi_{k}\right)}^{-1}P_{k}\\ P_{k}=(P_{k-1}(I-L_{k}\varphi_{k}^{T}))/\lambda_{k} \end{array}\right.$$
where $P_{k}$ is the covariance matrix, $K_{k}$ is the gain matrix, and $\lambda_{k}$ is the forgetting factor, which takes values in the range 0.9 < λ < 1, used to attenuate the influence of historical data on the current estimate.

Sensor noise and errors are inevitable in practical applications. When sensors produce abnormal errors, to enhance the robustness of FFRLS for estimating the total vehicle mass, M-estimation is integrated with FFRLS. The objective function is defined as follows:(16)$$J\left(\theta_{k}\right)=\rho \left(r_{k}\right)$$
where $r_{k}=(y_{k}-\varphi_{k}^{T}\theta_{k})$, function is defined as [27]:(17)$$\rho \left(r_{k}\right)=\left\{\begin{array}{cc} 0.5r_{k}^{2} & \left|r_{k}\right|\leq c\\ c|r_{k}|-0.5r_{k}^{2} & \left|r_{k}\right|>c \end{array}\right.$$

In this equation, c is an adjustable parameter, and the derivative of the performance function $J\left(\theta_{k}\right)$ with respect to c can be obtained as follows:(18)$$\frac{\partial J(\theta_{k})}{\partial \theta_{k}}=\dot{\rho }(r_{k})\frac{\partial r_{k}}{\partial x_{k}}=0$$

Definition $\psi_{k}=\frac{\dot{\rho }(r_{k})}{r_{k}}$, $\psi_{k}$ is the weight value, as shown in the following equation:(19)$$\psi_{k}=\left\{\begin{array}{cc} 1 & \left|r_{k}\right|\leq c\\ \frac{c}{\left|r_{k}\right|} & \left|r_{k}\right|>c \end{array}\right.$$

Introducing the weighting factor $\psi_{k}$ into the FFRLS, we obtain the Robust FFRLS, whose specific expression is:(20)$$\left\{\begin{array}{c} {\hat{\theta }}_{k}={\hat{\theta }}_{k-1}+K_{k}\psi_{k}(y_{k}-\varphi_{k}^{T}{\hat{\theta }}_{k-1})\\ K_{k}=P_{k-1}\varphi_{k}^{T}{\left(\lambda_{k}+\varphi_{k}^{T}P_{k-1}\varphi_{k}\right)}^{-1}P_{k}\\ P_{k}=(P_{k-1}(I-L_{k}\varphi_{k}^{T}))/\lambda_{k} \end{array}\right.$$

By introducing weight values $\psi_{k}$, when the new measurement value $r_{k}$ exceeds the set value, the influence of noise and abnormal measurement data in the sensor at moment *k* on the overall vehicle mass estimation is weakened by updating the weight values $\psi_{k}$, while also avoiding the influence of power fluctuations caused by gear shifts and braking on the overall vehicle mass estimation.

### 3.2. Mass Estimation Based on Robust EKF

The EKF algorithm is superior in dynamic estimation ability and adaptability to nonlinear systems, which can dynamically adjust the parameters in the state estimation process to adapt to different environmental and task requirements, ensuring the stability and reliability of state identification, thereby improving the accuracy and real-time performance of vehicle state parameter identification [28]. In addition, the EKF algorithm is favored for its capability to effectively handle nonlinear systems, such as tire characteristics and suspension dynamics, while maintaining superior real-time performance. This makes it particularly suitable for applications that require rapid response and accurate state estimation. By modeling and compensating for sensor noise and model uncertainties, the EKF mitigates error propagation and enhances the reliability of the results. Furthermore, it can efficiently manage multi-variable dynamic systems, such as acceleration and braking forces, without a significant increase in computational complexity or dimensionality. Compared to other algorithms like the Particle Filter (PF) and Unscented Kalman Filter (UKF), the EKF offers greater computational efficiency and ease of implementation in specific scenarios, especially when precise models are required for vehicle mass estimation. According to the longitudinal dynamic model established, the system’s state variables are set as shown in Equation (21):(21)$$x_{k}=(v_{k},\alpha_{k},M_{k})$$

In this equation, $v_{k}$ represents the vehicle’s speed at the current time, $\alpha_{k}$ represents the road slope angle at the current time, and $M_{k}$ represents the total vehicle mass at the current time.

When the vehicle is traveling steadily, the mass can be considered a constant. Assuming that the road slope angle, vehicle speed, and acceleration are continuous variables that do not undergo abrupt changes, the system prediction equation after discretizing over Δt time is:(22)$$\left\{\begin{array}{l} v_{k}=v_{k-1}+{\dot{v}}_{k-1}\Delta t+w_{1(k-1)}\\ \alpha_{k}=\alpha_{k-1}+w_{2(k-1)}\\ M_{k}=M_{k-1}+w_{3(k-1)} \end{array}\right.$$

In this equation, $w_{1(k-1)}$, $w_{2(k-1)}$, $w_{3(k-1)}$ represent the process noise at the previous time step, and ${\dot{v}}_{k-1}$ represents the previous acceleration. They can be expressed as follows:(23)$${\dot{v}}_{k-1}=\frac{1}{\delta }\left(\frac{T_{\left(k-1\right)}i_{g}i_{tr}\eta_{t}}{M_{k-1}r}-gf-\frac{C_{D}A\rho v_{k-1}^{2}}{2M_{k-1}}-g\alpha_{k-1}\right)$$

The system state prediction equation can be rewritten in matrix form as follows:(24)$$x_{k}=\left[\begin{array}{c} v_{k-1}+{\dot{v}}_{k-1}\Delta t\\ \alpha_{k-1}\\ m_{k-1} \end{array}\right]+w_{k-1}$$

The further system state transition equation can be expressed as:(25)$$x_{k}=f(u_{k-1},x_{k-1})+w_{k-1}$$
where $u_{k-1}$ is the control vector from the previous time step.

In the measurement update, the vehicle speed can be calculated using wheel speed sensors, and the slope angle can be measured using IMU, so the measurement equation of the system is:(26)$$z_{k}=Hx_{k}+o_{k}=\left[\begin{array}{ccc} 1 & 0 & 0\\ 0 & 1 & 0 \end{array}\right]x_{k}+o_{k}$$

In this equation, $z_{k}$ represents the measured value at the current time, H is the measurement matrix, and $o_{k}$ represents the measurement noise at the current time, which is assumed to be independent of $w_{k}$ and have a zero mean and a Gaussian white noise distribution.

Due to the strong nonlinearity of the system, the a priori estimate obtained by calculating the nonlinearization function *f* is not a unimodal Gaussian distribution, which does not meet the usage conditions of the EKF algorithm. A first-order Taylor series needs to be constructed to approximate the linear function. The first-order Taylor expansion at the previous estimate value ${\hat{x}}_{k-1}$ can be expressed as:(27)$$\begin{array}{c} x_{k}=f(u_{k-1},{\hat{x}}_{k-1})+F_{k-1}(u_{k-1},x_{k-1}-{\hat{x}}_{k-1})+w_{k-1} \end{array}$$

In this equation, ${\hat{x}}_{k-1}$ represents the estimated value of the system at time $x_{k-1}$, and $F_{k-1}$ represents the Jacobian matrix of the function *f* with respect to the state variables $x_{k-1}$, which can be expressed as:(28)$$F_{k-1}=\left[\begin{array}{ccc} 1-\frac{C_{D}A\rho v_{k-1}}{\delta M_{k-1}}\Delta t & -\frac{g}{\delta }\Delta t & \frac{C_{D}A\rho v_{k-1}^{2}r-2T(k-1)}{2\delta M_{k-1}^{2}r}\Delta t\\ 0 & 1 & 0\\ 0 & 0 & 1 \end{array}\right]$$

The prediction update equation for the system’s a priori estimate can be obtained as follows:(29)$${\hat{x}}_{k}^{-}=f(u_{k-1},{\hat{x}}_{k-1})$$

The covariance of the system’s prior estimate is:(30)$$P_{k}^{-}=F_{k-1}P_{k-1}F_{k-1}^{T}+Q$$
where ***Q*** is the covariance matrix of the process noise.

We further calculate the Kalman gain:(31)$$K_{k}=\frac{P_{k}^{-}H_{k}^{T}}{H_{k}P_{k}^{-}H_{k}^{T}+R_{k}}$$

Finally, the estimated value and error covariance are updated to obtain the final estimated value of the total vehicle mass:(32)$${\hat{x}}_{k}={\hat{x}}_{k}^{-}+K_{k}(z_{k}^{-}H_{k}{\hat{x}}_{k}^{-})$$(33)$$P_{k}=(I-K_{k}H_{k})P_{k}^{-}$$

The initial error covariance matrix ***P*_0_**, process noise ***Q***, and measurement noise ***R*** significantly influence the convergence speed and estimation accuracy of the EKF algorithm in practical applications. The diagonal elements of the error covariance matrix represent the variances of the state variables, while the off-diagonal elements denote the covariances between these variables. When variables are uncorrelated, the off-diagonal covariance terms should be set to zero. The initial error covariance reflects the uncertainty in the initial values of the state variables and is typically initialized as an identity matrix. Process noise ***Q*** is closely associated with model accuracy, the degree of model linearization, and discretization-induced errors. Measurement noise ***R*** depends on the characteristics and precision of the selected sensors. In this study, ***R*** is determined based on sensor measurement accuracy, and ***Q*** is adjusted to optimize the estimation performance. The vehicle mass estimation process using the EKF algorithm is illustrated in Figure 2.

### 3.3. Hybrid Architecture for Vehicle Mass Estimation Based on EKF and Robust FFRLS

The hybrid architecture for vehicle mass estimation based on EKF and Robust FFRLS is illustrated in Figure 3. Once the vehicle begins operation, both methods concurrently initiate the estimation of the vehicle’s total mass. The initial estimate from the Robust FFRLS algorithm serves as the input to initialize the EKF algorithm, thereby enhancing the accuracy of the EKF-based estimation. Ultimately, the *h* function effectively hybridizes the estimates derived from the Robust FFRLS and EKF algorithms to produce a stable and accurate mass output.

The *h* function is a function of vehicle speed, serving as a proportional factor to weigh the two methods with weight coefficients. The value of *h* is shown in Figure 4.

Upon the commencement of measurements, the Robust FFRLS-based method exhibits a shorter convergence time compared to the EKF-based method. Specifically, at lower vehicle speeds with small excitations, the Robust FFRLS-based method achieves higher estimation accuracy. Conversely, after a certain period of operation, the EKF-based algorithm provides more accurate estimates of the vehicle’s total mass at high speeds and under large excitations. The hybrid vehicle mass estimate based on the h function can be expressed as:(34)$$\hat{M}=h{\hat{M}}_{1}+(1-h){\hat{M}}_{2}$$
where ${\hat{M}}_{1}$ represents the vehicle mass value based on Robust FFRLS estimation, ${\hat{M}}_{2}$ represents the vehicle mass value based on EKF estimation, and $\hat{M}$ represents the final vehicle mass estimate output by the hybrid algorithm.

## 4. Results and Analysis

### 4.1. Introduction of Test Vehicle and Test Platform

To verify the effectiveness and accuracy of the proposed bus vehicle mass estimation, a bus model was built in TruckSim (2019.0), and the algorithm was validated through joint simulation between TruckSim (2019.0) and Matlab (2022b)/Simulink (2019.0). The basic parameters of the bus are shown in Table 1.

The bus model in TruckSim (2019.0) provides gear shift signals, brake signals, acceleration signals, speed signals, and engine torque signals for the algorithm, and also provides test conditions. The simulation frequency was set to 0.01. The noise and accuracy of the IMU sensor have a direct impact on the reliability of the measured data. In order to alleviate this problem, this paper uses multiple sampling and data smoothing techniques to filter the signal to improve the signal quality.

### 4.2. Vehicle Mass Estimation Test Under No-Load Conditions

To verify the accuracy and effectiveness of the vehicle’s total mass estimation under empty load conditions, the bus was configured as empty in Trucksim(2019.0) to simulate a passenger-free scenario. The experimental condition involved a straight road with an adhesion coefficient of 0.85 and a true mass of 7000 kg. Data collection began immediately upon vehicle startup and continued through various driving conditions, including acceleration, constant speed, and braking, to introduce external disturbances. The experimental results are presented in Figure 5a, the vehicle speed in Figure 5b, the acceleration in Figure 5c, and the engine output torque in Figure 5d.

As shown in Figure 5a, the Robust FFRLS method does not have an initial value, so the total vehicle mass starts from 0 and rises, and the sudden braking at the rising process delays the convergence speed. The convergence starts at 21 s with an initial estimation error greater than 10%, and the error is less than 10% after convergence. The two braking events at 115 s and 147 s will cause fluctuations in the estimated value of the total vehicle mass. The EKF algorithm has an initial value, which can quickly converge to the true value, and the estimated value after convergence has good stability and is not affected by sudden external stimuli such as braking. The RMSE of the three estimation methods is shown in Table 2, and the proposed hybrid algorithm combines the advantages of the two methods by allocating and fusing them, resulting in a smaller estimation error.

Compared to the half-load and full-load conditions, the RMSE of mass estimation under no-load conditions is higher. This is due to the fact that under no-load conditions, the engine torque required for the vehicle to reach the target speed is significantly lower, resulting in reduced input excitation to the system, which adversely affects estimation accuracy. Nonetheless, the overall error remains within an acceptable range.

Finally, from the perspective of computational efficiency, the hybrid algorithm exhibits lower computational efficiency compared to both the EKF and the Robust FFRLS. This reduced efficiency is attributed to the integration of two distinct algorithms within the hybrid approach, along with the introduction of varying weighting factors, which collectively increase computational complexity. The EKF algorithm also demonstrates lower computational efficiency relative to Robust FFRLS, primarily due to its more intricate calculation steps and processes. In practical applications, appropriately sacrificing computational efficiency to enhance the accuracy of vehicle mass estimation holds significant practical value for improving vehicle safety and energy-saving performance.

### 4.3. Vehicle Mass Estimation Test Under Half-Load Conditions

To verify the accuracy and effectiveness of the vehicle’s total mass estimation under half-load conditions, the bus was set to half-load in Trucksim (2019.0), simulating the vehicle’s passenger situation. The test conditions were kept consistent with the empty condition, with the actual vehicle mass being 10,000 kg. The test results are shown in Figure 6.

As shown in Figure 6, both Robust FFRLS and EKF algorithms can converge to the true value, and the convergence speed of EKF is generally faster than that of Robust FFRLS. When affected by sudden external stimuli such as braking, the influence of Robust FFRLS is much greater than that of EKF, and the estimated error of Robust FFRLS gradually increases later. The RMSE of the three estimation methods is shown in Table 2, and the proposed hybrid algorithm integrates the advantages of both algorithms by allocating and fusing them. The estimated value converges to the true value to a great extent, and the RMSE is 100.95.

In the half-load condition, the mass estimation accuracy of the vehicle is significantly higher compared to the no-load and full-load conditions, being approximately twice as accurate. This enhanced accuracy can be attributed to the greater excitation obtained by the system under half-load conditions, which facilitates more precise estimations.

Finally, the computational efficiency of the method under above condition proposed in this paper is similar to that of the EKF algorithm, but lower than that of the FFRLS. However, under the half-load condition, the hybrid algorithm has the smallest RMSE among the three types of loads and achieves the best estimation effect.

### 4.4. Vehicle Mass Estimation Test Under Full-Load Conditions

To verify the vehicle’s overall mass estimation in the full load condition, the bus was set to full load in Trucksim (2019.0), simulating the vehicle reaching its maximum load capacity. The test conditions were kept consistent with the empty load, with the actual mass being 13,000 kg. The test results are shown in Figure 7.

As shown in Figure 7, the convergence rate of Robust FFRLS is much lower than that of the EKF algorithm, and the EKF algorithm demonstrates higher estimation accuracy and stability. The RMSE of the three estimation methods is shown in Table 2, and the proposed hybrid algorithm integrates the advantages of both methods by allocating and fusing them. The estimated value converges to the true value in the vicinity, with an RMSE of 228.91, which is smaller than the other two methods.

Under full load conditions, the accuracy of mass estimation is lower compared to that under half load. This reduction in accuracy can primarily be attributed to expanded model uncertainty, and more pronounced nonlinear effects within the system. Consequently, these factors diminish the effectiveness of the EKF algorithm in managing complex dynamics, which in turn impacts the precision of mass estimation.

Finally, under full-load conditions, despite the proposed method exhibiting lower computational efficiency, it achieves superior estimation accuracy, thus guaranteeing the active safety function and optimal braking force distribution of the vehicle. Furthermore, in practical applications, advancements in intelligent chassis technology can promote the integration of high-performance chips into chassis domain controllers, thereby significantly enhancing the computational efficiency of the proposed method.

## 5. Conclusions

This paper presents a hybrid estimation method for the bus mass, which integrates the estimated masses from Robust FFRLS and EKF-based methods using a proportional coordination factor. Specifically, by incorporating M-estimation with FFRLS, the algorithm enhances recognition accuracy during braking and gear-shifting conditions. The hybrid algorithm leverages the strengths of both estimation methods, enabling rapid convergence to the true value and ensuring stable estimation results. Simulation results demonstrate that, under three loading scenarios—empty, half-loaded, and fully loaded—the proposed algorithm exhibits smaller RMSE compared to single algorithms, thereby achieving superior estimation accuracy and stability.

Due to the constraints imposed by experimental conditions, the verification phase of this study was limited to simulation experiments. For future research, algorithm validation will be conducted in real-world environments using actual vehicle platforms. Moreover, lateral motion dynamics will be incorporated into vehicle mass estimation, leading to the development of a comprehensive estimator that integrates both longitudinal and lateral dynamic model.

## Figures and Tables

**Figure 1 sensors-25-01741-f001:**
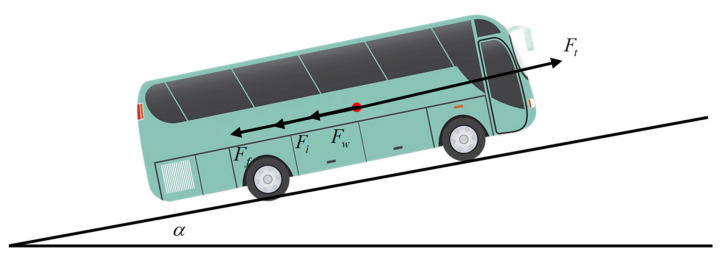
Bus Longitudinal Dynamics Model.

**Figure 2 sensors-25-01741-f002:**
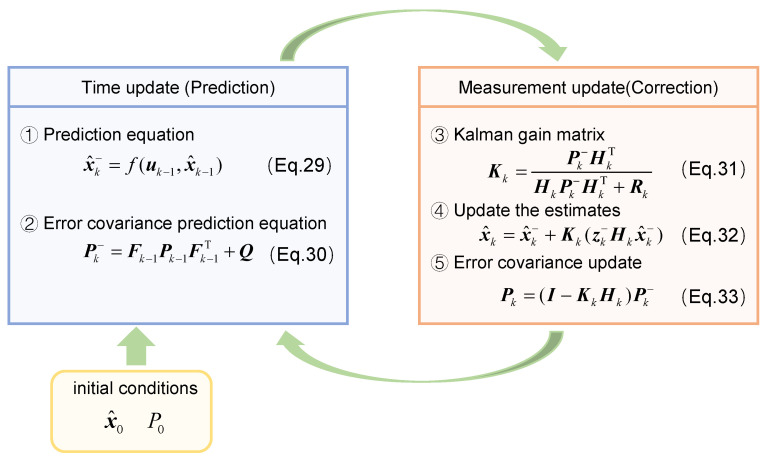
Vehicle mass estimation process based on EKF algorithm.

**Figure 3 sensors-25-01741-f003:**
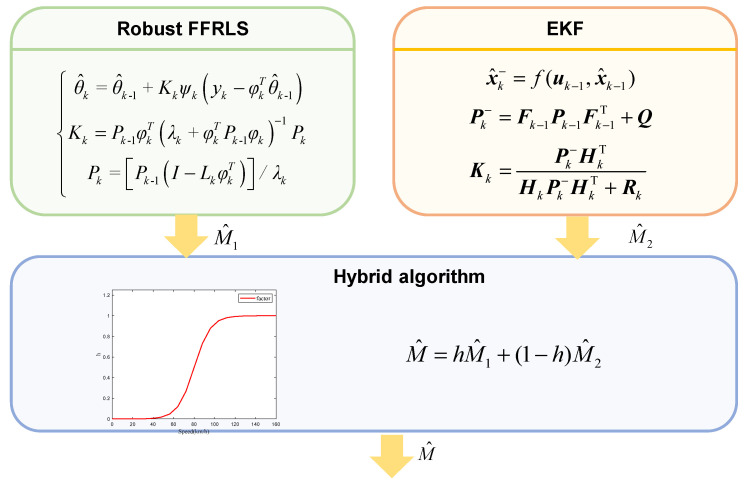
Hybrid algorithm architecture for vehicle mass estimation.

**Figure 4 sensors-25-01741-f004:**
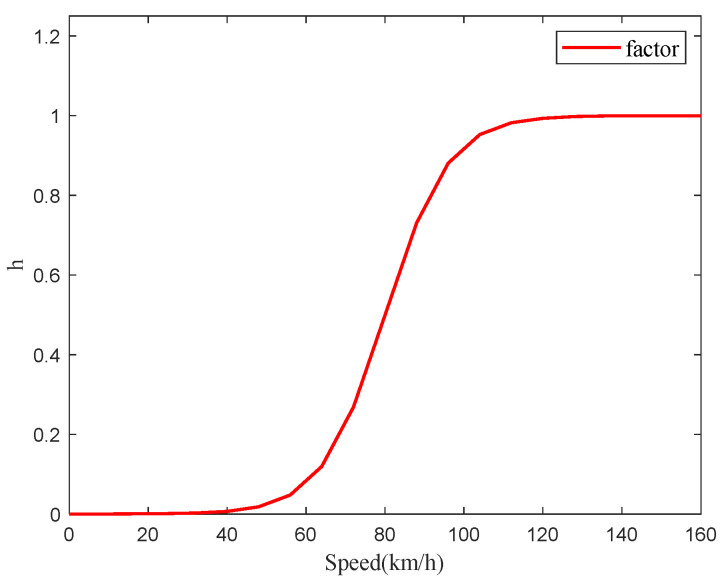
The process of proportional coordination factor with speed.

**Figure 5 sensors-25-01741-f005:**
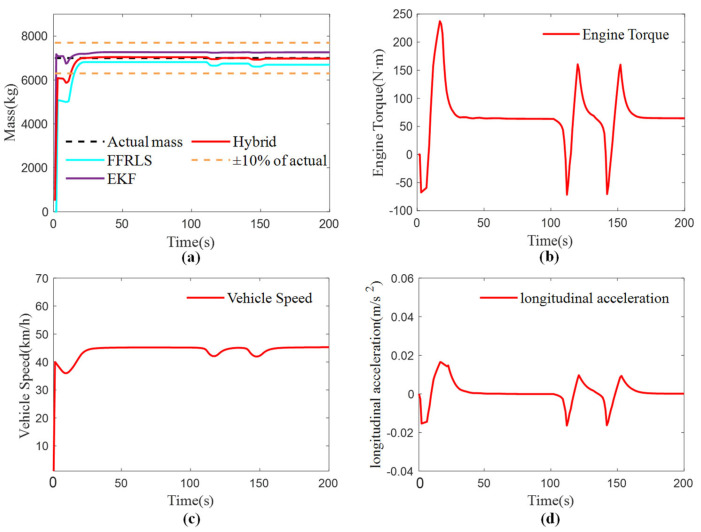
The results of vehicle mass estimation under no-load condition: (**a**) Vehicle Mass Estimation Results; (**b**) Engine torque; (**c**) Vehicle speed; (**d**) Longitudinal acceleration.

**Figure 6 sensors-25-01741-f006:**
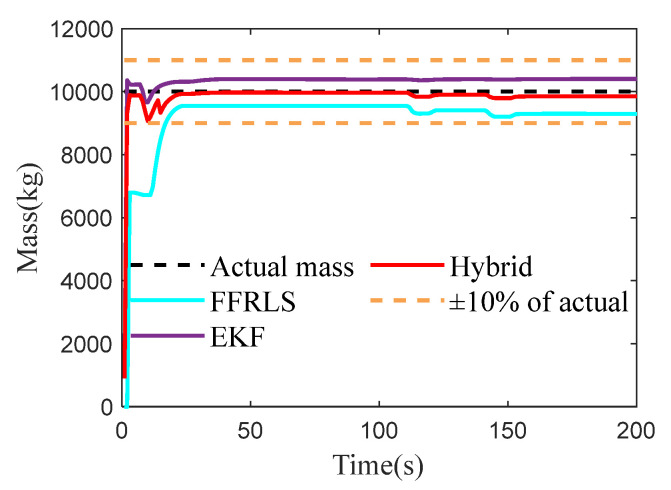
The results of vehicle mass estimation under half-load condition.

**Figure 7 sensors-25-01741-f007:**
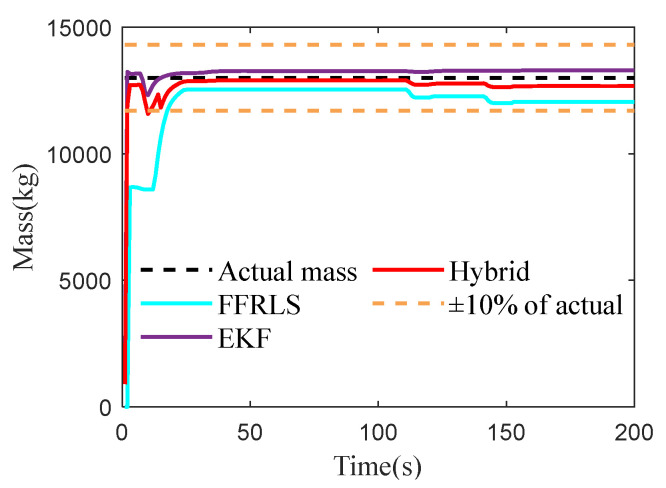
The results of vehicle mass estimation under full-load conditions.

**Table 1 sensors-25-01741-t001:** Basic parameters of the test.

Parameters	Unit	Value
Vehicle mass (full-load)	kg	13,000
Vehicle mass (half-load)	kg	10,000
Vehicle mass (no-load)	kg	7000
Coefficient of air resistance *C_D_*	-	0.69
Windward area *A*	m/s^2^	7.9
Air density *ρ*	Kg/m^3^	1.206
Effective tire radius *r*	m	0.49
Coefficients of rolling resistance *f*		0.0005
Axle ratio *i*_0_	-	3.545
Mechanical efficiency *η*	-	0.95
Forgetting factor *λ*	-	0.999

**Table 2 sensors-25-01741-t002:** RMSE of different methods in no-load, half-load, and full-load state.

Condition	Methods	RMSE (kg)	Computational Efficiency (s)
no-load	Robust FFRLS	324.73	9.2
	EKF	299.26	12.3
	Hybrid algorithm	273.84	12.5
half-load	Robust FFRLS	589.27	9.0
	EKF	389.16	13.1
	Hybrid algorithm	100.95	13.6
full-load	Robust FFRLS	718.43	9.4
	EKF	275.46	12.9
	Hybrid algorithm	228.91	13.7

## Data Availability

The raw data supporting the conclusions of this article will be made available by the authors on request.

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
