# Peer review of "Estimating Bus Mass Using a Hybrid Approach: Integrating Forgetting Factor Recursive Least Squares with the Extended Kalman Filter"

_sensors, 2025, doi:10.3390/s25061741_

Round 1
Reviewer 1 Report
Comments and Suggestions for Authors
In the current study, the authors propose a hybrid algorithm for estimating vehicle mass using robust FFRLS and EKF methods. Their approach combines two independent estimations, weighted by a proportional coordination factor, to improve accuracy. The method utilizes data from existing CAN-bus and IMU sensors, eliminating the need for additional hardware. They evaluated the algorithm through MATLAB-TruckSim simulations under empty, half-load, and full-load conditions. Results showed an error rate below 10%, outperforming single-method approaches in accuracy, safety, and stability. While the study was interesting, the following are issues for the consideration of the authors for the various aspects of the manuscript:
Abstract:
The abstract presented a strong foundation with a well-organized discussion of existing methods and their limitations. There was also a clear motivation for the hybrid approach.
However, minor refinements in sentence structure, grammar, and clarity would improve readability. Further, explicitly stating the research gap and why buses were chosen would enhance justification. Finally, information about experimental validation beyond simulations could be provided in the abstract and the article in addition to future work or real-world testing could strengthen the practical significance of the abstract in my considered opinion.
Introduction:
The introduction section is well-structured, technically sound, and provides a thorough literature review. However, this section needs clarity, a more explicit research gap statement and minor refinements in sentence structure and formatting.
Methods:
Overall, the methods section is well-structured and theoretically sound, offering a solid foundation for bus vehicle modeling and mass estimation. However, additional experimental validation and discussion of practical limitations would enhance its applicability and robustness.
Results and discussion:
The results and discussion section provides a strong and structured analysis of the proposed hybrid estimation method. It convincingly demonstrates its superiority over standalone FFRLS and EKF approaches. However, the paper would benefit from discussing computational efficiency (that is, processing speed of computer used for simulations and time required to run simulations), real-world applicability, and additional dynamic test scenarios to enhance the robustness of findings.
Conclusion:
The conclusion effectively summarizes the study’s contributions but could be improved by acknowledging limitations and suggesting future research directions. Adding these elements would provide a more balanced and forward-looking perspective, strengthening the paper’s appeal to the readership and impact.
Comments on the Quality of English LanguageIn the current study, the authors propose a hybrid algorithm for estimating vehicle mass using robust FFRLS and EKF methods. Their approach combines two independent estimations, weighted by a proportional coordination factor, to improve accuracy. The method utilizes data from existing CAN-bus and IMU sensors, eliminating the need for additional hardware. They evaluated the algorithm through MATLAB-TruckSim simulations under empty, half-load, and full-load conditions. Results showed an error rate below 10%, outperforming single-method approaches in accuracy, safety, and stability. While the study was interesting, the following are issues for the consideration of the authors for the various aspects of the manuscript:
Abstract:
The abstract presented a strong foundation with a well-organized discussion of existing methods and their limitations. There was also a clear motivation for the hybrid approach.
However, minor refinements in sentence structure, grammar, and clarity would improve readability. Further, explicitly stating the research gap and why buses were chosen would enhance justification. Finally, information about experimental validation beyond simulations could be provided in the abstract and the article in addition to future work or real-world testing could strengthen the practical significance of the abstract in my considered opinion.
Introduction:
The introduction section is well-structured, technically sound, and provides a thorough literature review. However, this section needs clarity, a more explicit research gap statement and minor refinements in sentence structure and formatting.
Methods:
Overall, the methods section is well-structured and theoretically sound, offering a solid foundation for bus vehicle modeling and mass estimation. However, additional experimental validation and discussion of practical limitations would enhance its applicability and robustness.
Results and discussion:
The results and discussion section provides a strong and structured analysis of the proposed hybrid estimation method. It convincingly demonstrates its superiority over standalone FFRLS and EKF approaches. However, the paper would benefit from discussing computational efficiency (that is, processing speed of computer used for simulations and time required to run simulations), real-world applicability, and additional dynamic test scenarios to enhance the robustness of findings.
Conclusion:
The conclusion effectively summarizes the study’s contributions but could be improved by acknowledging limitations and suggesting future research directions. Adding these elements would provide a more balanced and forward-looking perspective, strengthening the paper’s appeal to the readership and impact.
Reviewer 2 Report
Comments and Suggestions for Authors
The article "Estimating Bus Mass Using a Hybrid Approach: Integrating Forgetting Factor Recursive Least Squares with the Extended Kalman Filter" is well written. Paying attention to the following comments can improve the quality of the research:
1- Provide references for all paragraphs of the Introduction. Some, especially the final paragraphs, lack references.
2- Re-examine equation (1), it seems that there is an error in it.
3- Explain the algorithm using the EKF algorithm further.
4- In Figure 5, all axes should be the same. The format should be followed
5- Merge Tables 2 and 3 for a more detailed review and explain the differences
Reviewer 3 Report
Comments and Suggestions for Authors
This paper proposed a hybrid approach for vehicle mass estimation, which combines FFRLS and EKF. Although interesting studies are presented by the authors, the reviewer has some concerns (both minor and major) about this research. The details are summarized as follows.
- Please show the full name of FFRLS and EKF in the abstract.
- Lines 42-44. Can the authors provide some narratives on why tire pressure and suspension position will help estimate the vehicle mass?
- Lines 81-86. Remove since they are summarized again in a later paragraph; see Lines 110-124.
- Need to proofread the manuscript before submission. Use professional English expressions instead of direct translation from another language. For example, the expressions like “fusion”, “fuses”, “fused” are everywhere in the text. They confused the readers a lot. Replace them with words like “hybrid”, “combines”, etc.
- Line 141. Move section title to next page.
- Equation 1. Is this a dynamic equation? Where does the acceleration term show up?
- Lines 169-171 including Equation 5. Not logical. Why does “v” still show up in the equation while it is said the impact of “v” is negligible?
- Equation 8. Later in the paper, the authors treated mass as a state variable (meaning it’s time-varying), which invalidates this equation where M stays constant. How would you explain such discrepancy? Does your method only work for constant mass case?
- Equations 9-11. How are these equations used in the model? Why only yaw rate is considered? What about pitch and roll?
- Equation 14. How to obtain v_x and alpha? What are the values for T_tq and i_s? What about f_0 and f_1? These are all unclear to the readers.
- Please avoid splitting a single word in different lines too frequently. It affects the readability of the paper.
- Figure 4 with Equation 34. It appears to the reviewer that this is the only new contribution of this work to this research field. However, based on what the weight profile is generated? There’s no quantitative study. The authors explained qualitatively that FFRLS works better for low-speed cases while EKF works better for high-speed ones. Is this really in line with Figure 5(a)(c)?
- All figures in the results section. Is +-10 supposed to mean +-10%?
- Since you are using sensors and algorithms to estimate the mass, what is the uncertainty of such mass calculation and how did you analyze it? It turns out to be the most interested value since it indicates the trustworthiness of your method.
- The case study only provides the estimation of a constant mass. Again, will the same method work for time-varying mass with the same performance?
The quality of the paper has some room to be improved considering the above-mentioned concerns. Major revision and addressment of all the comments should be required before this paper can be recommended for publication.
Comments on the Quality of English LanguageIncluded in the comments.
Round 2
Reviewer 2 Report
Comments and Suggestions for Authors
The authors have responded to the comments.
Reviewer 3 Report
Comments and Suggestions for Authors
My concerns have been addressed by the authors.